# Fucoxanthin Ameliorates Atopic Dermatitis Symptoms by Regulating Keratinocytes and Regulatory Innate Lymphoid Cells

**DOI:** 10.3390/ijms21062180

**Published:** 2020-03-22

**Authors:** Chika Natsume, Nao Aoki, Tomoko Aoyama, Keisuke Senda, Mio Matsui, Airi Ikegami, Kosuke Tanaka, Yasu-Taka Azuma, Takashi Fujita

**Affiliations:** 1Molecular Toxicology Lab., Ritsumeikan University, Shiga 525-8577, Japan; chika.ntsm@gmail.com (C.N.); ph0075eh@ed.ritsumei.ac.jp (N.A.); 723tomoko@gmail.com (T.A.); casebykeisuke84@yahoo.co.jp (K.S.); sb0006vi@gmail.com (M.M.); ph0091fv@ed.ritsumei.ac.jp (A.I.); ko.tanaka@kobayashi.co.jp (K.T.); 2Laboratory of Veterinary Pharmacology, Division of Veterinary Science, Osaka Prefecture University, Graduate School of Life and Environmental Science, Izumisano, Osaka 598-8531, Japan; azuma@vet.osakafu-u.ac.jp

**Keywords:** Fucoxanthin, tacrolimus, atopic dermatitis, keratinocytes, eosinophil, GATA transcription factors, Il-2, Il-10, TGF-β, mast cells, ILCreg, ILC2

## Abstract

Fucoxanthin (FX) is a xanthophyll that is contained abundantly in marine plants. The biological action of FX includes its antioxidant and anti-lipogenic activities, while the precise action of its mechanisms on skin cells has not yet been clarified. The current study examined the effect of FX in comparison with tacrolimus (TAC) on NC/Nga mice, which are an atopic dermatitis (AD) model. FX topical treatment dramatically ameliorated itching behavior over the TAC treatment, which was insufficient for improvement of AD symptoms. In Nc/Nga mice, FX or TAC applied to the skin inhibited eosinophil infiltration with decreased expression of Il-33. FX also stimulated Il-2, Il-5, Il-13, Il-10, and TGF-β expression levels, and Sca1^+^Il-10^+^TGF-β^+^ regulatory innate lymphoid cells (ILCreg) were dominantly observed in FX treated skin epidermal keratinocytes and dermal layers. This combined evidence demonstrated that FX exerts anti-inflammatory effects on keratinocytes and ameliorates AD symptoms by regulating ILCreg to normalize immune responses in an atopic dermatitis model.

## 1. Introduction

Atopic dermatitis (AD), which affects at least 15% of children in developed countries, causes dry, scaly, and itchy skin [1]. A complex interaction of genetic predisposition, environmental exposures, and immunologic mechanisms drives the development of AD [2,3]. Pruritus significantly lowers the quality of life in AD pathology [4]. Allergic condition recruits mast cell precursor cells from the blood vessels to the dermis, and the maturated mast cells in the dermis play a central role in itch development [5]. Mature mast cells express molecules such as FcεRI and histidine decarboxylase (HDC). Elevated IgE causes degranulation by binding to a high-affinity receptor FcεRI expressed on mast cells, resulting in itching in allergic diseases [6,7]. Subsequently, itching is triggered by chemical mediators including histamine, proteoglycans, and neutral proteases released from the matured mast cells. Chemical mediators are transmitted to natriuretic polypeptide b and Mas-related G protein-coupled receptor a3 (Mrgpra3) positive sensory neurons, and itching behavior occurs [8].

Mast cell function is regulated indirectly by non-mast cells, including adaptive helper T cells [6,9]. Immunosuppressant treatment for AD using a calcineurin inhibitor, tacrolimus (TAC), broadly inhibits their function [10]. Recent studies have demonstrated that innate lymphoid cells (ILCs), which are distinct from adaptive helper T cells, are localized in the skin, and the function of ILCs underlies the pathogenesis of AD [11]. Therefore, compounds that modulate mast cells and adaptive helper T cells would require a systematic reassessment of immune function.

ILCs are currently classified into at least three different populations based on developmental requirements, transcription factor expression profiles, and/or effector cytokine expressions [12]. However, the various functional properties of skin ILCs are poorly defined. For example, CD4 positive Th2-type helper T cells and regulatory T cells (Treg) are characterized by the expression of GATA3 and FoxP3, respectively, while CD4 negative ILC2 produces Th2 cytokines (Il-5 and Il-13). Its subpopulation, ILC2_10_ in the lung (also called ILCreg in the intestine), expresses GATA3 capable of producing Th2 cytokines and Il-10 in the lung [13,14]. In the skin, Treg regulates the differentiation of epithelial stem cells around hair follicles and exerts a different localization and function from Treg in systemic circulation [15]. Therefore, skin immunity needs to be reconsidered from the viewpoint of regulating ILCs to overcome multiple skin diseases.

So far, over 850 carotenoids have been found in nature. Some carotenoids, such as β-cryptoxanthin, lutein, astaxanthin, fucoxanthin (FX), lycopene, and β-carotene, have physiological functions, including lifestyle-related disease prevention and carcinogenesis prevention [16]. Essentially, the lipophilic properties of carotenoids promise easy distribution in adipose tissue and facilitate suppressive effects on lipid accumulation [17]. Apart from the lipid metabolism, several carotenoids inhibit the degranulation of mast cells by suppressing the antigen-induced aggregation of the high affinity IgE receptor (FcϵRI) in vitro [18,19]. Additionally, FX ameliorates the decrease in filaggrin (Flg) expression under the condition of UV-A irradiation and sunburn symptoms that are independent of antioxidant activity [20]. Thus, the antioxidant activity-independent mechanisms of FX might be unique properties that could enhance the understanding of the biological importance among carotenoids.

In our current study, the application of FX to the skin provided anti-inflammatory effects on keratinocytes. FX suppressed expression of keratinocyte-derived Il-33, suppressed eosinophil recruitments, and stimulated the expression of TGF-β, which is required for tissue remodeling. FX also temporarily suppressed mast cell-dependent scratching behavior. These effects were common with TAC and contributed to improving AD symptoms. However, TAC had no effect on the expression of ILCreg-derived anti-inflammatory cytokines such as Il-2 and Il-10. The different effects on ILCreg could explain for why FX is more effective at improving AD symptoms than TAC. We report here that FX brought about a dramatic improvement in AD symptoms by regulating ILCreg.

## 2. Results

### 2.1. FX Ameliorated AD Symptoms in Nc/Nga Mice

When each material was externally applied to NC/Nga mice skin exhibiting a mild phenotype, 0.1% FX significantly improved AD pathology with better reproducibility than 0.1% TAC (Figure 1A). Vaseline (Vas) treated skin showed a clot from the occipital region to around the scapula due to a scratch, and TAC resulted in improved skin symptoms over the Vas treated group. From the observation of mice, the scratching behavior was clearly suppressed by the treatment of FX or TAC (Figure 1B). Until day 15, the scratching behavior was reduced by FX or TAC, but increased again on day 35. Although the improvement effect was also recognized by TAC ointment, the improvement effect of FX was significant. From the follow-up analysis of trans-epidermal water loss (TEWL) values over time, transcutaneous water evaporation above 40 g/m^2^·hour was recorded in all mice (Figure 1C). In TAC treated skin, erythema/hemorrhage and excoriation/erosion were observed, but in the FX treated group, no bleeding or blood clotting was observed (Figure 1A,D). FX significantly reduced the modified clinical skin severity score from day 22 after the start of application. During experiments, elevated serum IgE values in Vas-, FX-, or TAC-treated mice were not changed, respectively (Appendix A). There was no effect on lymphocytes proliferation (Appendix A). Next, the appearance of mast cells in the affected area was examined by tissue histological analysis (Figure 2A). As for FX or TAC, the number of toluidine blue (TB)-positive mast cells was significantly reduced as compared with the Vas treated group (Figure 2B). Pathological analysis revealed no epidermal thickening after five weeks of continuous application of each compound. qRT-PCR analysis showed that GATA1 and GATA2 were regulated by FX or TAC in vivo (Figure 2C). FX or TAC significantly inhibited FcεRlα, but mature granule marker histidine decarboxylase (HDC) tended to be upregulated by TAC. The detailed analysis was conducted in a mast cell differentiation model using bone marrow-derived mast cells (BMMCs) (Figure 2D,E). Condition medium of WEHI-3 (WHVI-3-CM) for three weeks induced TB-positive cells, but they were suppressed by FX or TAC in a concentration-dependent manner (Figure 2D). The appearance of mast cell tryptase positive was suppressed by FX or TAC (Figure 2E). After differentiation for three weeks, FX showed concentration-dependent inhibitory effects on degranulation of BMMCs (Appendix A).

### 2.2. Keratinocyte-Mediated Signal Regulated by FX and TAC

As shown in Figure 1B, FX had an effect of suppressing itch more effectively than TAC, but examination of its effect on mast cell differentiation did not reveal a significant difference between the effects of FX and TAC. Next, we examined the expression levels of keratinocytes functions (Figure 3). Epithelial cell-derived factors produced from the epithelium by allergen stimulation are eosinophils, basophils, and mast cells that are not mediated by Th2 cells [9,11]. Il-33 expression levels were downregulated in FX or TAC treated skin to a similar extent (Figure 3A). Thymic stromal lymphopoietin (TSLP) expression levels were not influenced by FX or TAC, while TSLP receptor (TSLPR) expression levels were downregulated by FX or TAC to a similar extent. FX or TAC did not influence inflammatory cytokine expression levels (Appendix A), NF-κB activities (Appendix A), or cell viabilities in human keratinocytes HaCaT cells (Appendix A). IL-33 was known as an inducer of eosinophil recruitment [14]. We then examined the effect of FX and TAC on eosinophils infiltration in mice (Figure 3B). As expected, eosinophils with deeply stained granules were often found only in Vas treated skin.

We focused on lymphoid cells, as FX and TAC exerted an equivalent inhibitory effect on mast cells and keratinocytes in itch suppression. FX but not TAC dramatically stimulated Il-2 and Il-10 expression levels in Nc/Nga mice (Figure 4A). Not shown in the data, Il-4 expression levels were not detectable (Cq value > 40). FX significantly enhanced the expression of Il-5 and Il-13. Il-17 expression levels were significantly suppressed by TAC. TGF-β expression levels were dramatically upregulated by each compound.

In an attempt to determine the cell markers, we performed immunohistochemical analysis on skin sections to examine the effects of FX on ILCreg (Figure 4B). Sca1—a well-known stem cell marker—as well as ILCs and Sca1^+^Il-10^+^ cells were all classified as ILCreg [14]. As a result of the examination, Il-10^+^TGF-β^+^Sca1^+^ ILCreg were dominantly observed in FX treated groups in dermis compared to Vas or TAC treated groups. Il-10^+^ cells were detected in the epidermis and dermis of FX treated skin. Compared with FX treated skin, Il-10 immunoreactivity in TAC treated skin was diminished. Sca1 immunoreactivity in Vas was weaker than FX. In addition, TGF-β^+^ cells were abundantly observed in the epidermis of the FX and TAC treated groups. In immunohistochemical studies to determine Il-2-producing and GATA3^+^ cells, we observed that almost all Sca1^+^ILCreg expressed Il-2 and GATA3 (Figure 5A). In the FX-treated group, Il-2^+^GATA3^+^ cells were colocalized in edge of Sca1^+^epidermis and Sca1^+^ILCreg. GATA3 and Il-2 expressions were inversely correlated (GATA3^high^Il-2^low^ or GATA3 ^low^Il-2^high^) and GATA3 and Sca1 expressions were well-correlated in the epidermis (GATA3^high^Sca1^high^). In the TAC treated group, GATA3^+^ cells in the epidermis were broadly observed in the epidermis and Il-2 negative ILC2. GATA3 positive cells in the dermis were merged with Sca1^+^ ILCreg. Finally, we examined ILC2 localization (Figure 5B). ST2 known as Il-33R is a marker for inflammatory ILC2 [21]. Il-10^-^Il-5^+^ST2^+^ ILC2 were concentrated in Vas treated dermis nearby epidermis, whereas Il-10^+^Il-5^+^ST2^-^ ILCreg were abundant in FX treated groups. Il-10^+^Il-5^+^ST2^-^ ILCreg were few in TAC treated group. Pulmonary ILC2_10_ or intestinal ILCreg expressed CD45R and CD25 [13,14]. In FX treated group, almost all Il-2^+^ cells were co-expressed CD45R and CD25, while CD45R^-^CD25^-^Il-2^+^ cells were observed in the TAC treated group (Figure 6), indicating that FX modulates different populations from TAC-regulating Il-2 producers. These combined results demonstrated that FX stimulates Il-2^+^Il-5^+^Il-10^+^CD25^+^CD45R^+^Sca1^+^ ILCreg in AD skin.

## 3. Discussion

Atopic dermatitis is a common chronic skin disease that begins early in life and can adversely affect the quality of life of the patient. Optimal skin care practices and topical corticosteroids are still the cornerstone of treatment for this disease. Treatment of AD should be directed to skin barrier recovery, including skin hydration and skin repair, reduction of itching, and, if necessary, reduction of inflammation. Therefore, successful management of AD requires an approach that includes patient and career education, optimal skin care practices, anti-inflammatory treatment, and skin treatment with topical corticosteroids and topical calcineurin inhibitors. It is effective to suppress the maturation of mast cells to reduce itching, and we proceeded with research from this point of view. Unexpectedly, FX had an improvement effect on AD skin symptoms superior to TAC. In addition to mast cell maturation, we anticipated the existence of important mechanisms leading to novel AD improvement effects.

Our results showed that fucoxanthin regulates ILCreg subsets in addition to its effect on mast cells. FX and TAC reduced TB-positive mast cells, while their actions did not completely abolish mast cells (Figure 2A, B). Scratching behavior decreased until day 15, but reversed again until day 35 (Figure 2D). As for the modified clinical skin severity score, no significant improvement effect was observed on day 15, and FX showed an improved effect from the initial skin symptoms on day 22. The major difference from TAC was the apparent reduction in erythema/hemorrhage. These results suggest that FX results in improved skin symptoms through mast cell-independent mechanisms. For this reason, we focused on the analysis after five weeks of application where FX showed improvement in symptoms.

Early mast cell activities and scratching behavior seemed to be correlated. Mast cell differentiation is controlled by several master transcription factors [22]. GATA2 mRNA was clearly detected in most mast cell lines, but GATA1 mRNA expression was inactivated in some types of mast cell lines and was expressed in mature mast cells [23,24,25]. Thus, our data indicated that FX, like TAC, suppressed the mast cell differentiation processes in isolated of BMMCs (Figure 2D,E). There are two major differences between the effects of FX and TAC on AD symptoms. One is the difference in FcεRI sensitivity of mast cells that have already infiltrated into the dermis. Signaling factors that control FcεRI sensitivity, such as Mrgprx2, affect mast cell function [26]. Mrgpr family genes expressed in sensory neurons are also expressed in mast cells [27]. In skin mast cells, the expression of the Mrgpr family was characterized by high expression of type b8 and b13 and low expression of Mrgprx2. Some carotenoids convert to retinoic acid and attenuate degranulation of mast cells via Mrgprx2 in vivo [28,29]. Another is the role of mast cells in the microenvironment. Mast cell knockout (Kit^W–sh^/^W–sh^) mice easily develop severe AD-like skin inflammation [30]. Conversely, recent examinations demonstrate that mast cells show the potential for restoring homeostasis [31]. For instance, Il-10 produced by mast cells expand mast cells by autocrine and limit leukocyte infiltration and epidermal hyperplasia [32,33,34], indicating the protective role of mast cells against allergy. In contrast to the suppressive effects of FX or TAC in BMMCs, the decreased scratching action by FX or TAC became active again (Figure 1C). The number of TB-positive cells was only about 40% lower than the Vas treated group (Figure 2A,B). It is therefore possible that the number of mast cells was increased in vivo. Nevertheless, because of the significant differences in the effects of FX and TAC on AD symptoms (Figure 1D), we hypothesized that FX would improve AD symptoms by orchestrating immune cells including other cell types.

Bottleneck of itching behaviors in AD seems to occur not only in mast cells, but also in cell-to-cell communication between mast cells and other cell types including eosinophil, keratinocytes, and ILCreg. FX and TAC has exhibited anti-inflammatory effects via keratinocytes (Figure 3). Additionally, FX or TAC stimulated TGF-β expression levels in keratinocytes (Figure 4). These results suggest that TAC mainly improved AD symptoms by regulating keratinocytes, which directly support anti-inflammatory and immunosuppressive responses that are important for reestablishing skin homeostasis [35]. However, the weak healing effect of TAC on AD symptoms could be due to lack of ILCreg-derived cytokines.

Sca1 is a marker for a wide range of stem cells. Sca1 positive cells were observed in almost all areas of the skin, but ILCreg, which produces Il-10, showed a strong positive reaction to Sca1. Interestingly, GATA3 expression in the granular layer of the epidermis correlated with Sca1 expression, but it was inversely correlated with Il-2 expression in the FX treated group (Figure 5A). GATA3 expression patterns were not influenced by FX or TAC, whereas GATA3^high^ cells in the epidermis were frequently detected as Sca1^high^ cells in the FX treated group, suggesting that FX preferentially recruits ILCreg showing weak expression of Sca1 and GATA3 in the skin. Although GATA3 in keratinocytes contributes to cell differentiation and the expression of barrier proteins, GATA3 expression is reduced in both inflammatory skin diseases of AD and psoriasis [36,37,38,39,40]. The decreased GATA3 levels in keratinocytes contributes low Flg expression in two pathological conditions, but the relationship between mature Flg expression and its regulation after transcription is not fully understood. Further investigation will be required.

ILC2 also contributes to the functions of mast cells [41,42]. Recent reports demonstrated that Il-10-secreting pulmonary ILC2_10_ expresses GATA3 and produces Il-5 and Il-13 [14]. Il-10-secreting intestinal ILCreg also secrets TGF-β, which amplifies ILCreg expansion by autocrine, and exerts anti-inflammatory function during intestinal inflammation [13,43]. FX stimulates Il-2, Il-5, Il-13, Il-10, and TGF-β (Figure 4A), and Sca1^+^Il-10^+^TGF-β^+^ cells were abundantly observed in FX treated skin (Figure 4B). GATA3^low^ cells showed Il-2^high^ (Figure 5A), and Il-5^+^Il-10^+^ cells showed ST2 negative in the FX treated skin (Figure 5B). This suggests that FX dominates ILCreg rather than inflammatory ILC2. Additionally, the cells producing Il-10 were correlated with CD45R and CD25 expression in TX treated skin (Figure 6), suggesting that CD45R and CD25 are useful markers for skin ILCreg. A previous report by Hsieh et al. [44] demonstrated that IL-2 therapy is effective for AD patients. Il-2 functions as an elicitor for ILC2 expansion in the lungs and intestines [45,46]. The direct effect of FX on ILCreg characterized by the production of Il-2, Il-10 and TGF-β is very likely to be stronger than TAC, and Il-2 might exert to expand ILCreg by autocrine in the skin. Additionally, Il-2 expression levels in pulmonary ILC2_10_ were weaker than ILC2, and GATA3 expression levels were similar between ILC2_10_ and ILC2 [14] and intestinal ILCreg lacks Il-2 production and GATA3 [13]. Therefore, skin ILCreg had different properties from pulmonary ILC2_10_ and intestinal ILCreg. TGF-β expression in Vas-treated skin decreased in the epidermal layer (Figure 6). AD was also correlated with reduced TGF-β production, with TGF-β deficiency showing a severe inflammation phenotype [47,48]. Therefore, keratinocytes and ILCreg regulated by FX could contribute to ameliorating AD symptoms in our experimental model. GATA3 inhibitor ameliorates Th2 allergy symptoms including asthma and AD [49]. However, the effects of direct GATA3 inhibition on skin ILCs must be considered again.

We propose here that FX would help to improve AD symptoms more effectively by regulating ILCreg in the skin. The safety of FX was confirmed in a three-dimensional skin model [50]. Although our results may not currently be extrapolatable to humans, they still show that FX has the potential to become a new option for AD prevention and treatment.

## 4. Materials and Methods

### 4.1. Fucoxanthin Purification and Materials

FX (purity: 70%) was purchased from Phytolox (Okinawa, Japan) and purified as described previously [20] with several modifications. Briefly, the powder containing fucoxanthin was dissolved in acetone, then separated by over silica gel column chromatography (Wakosil C-200, WAKO Pure Chemicals, Osaka, Japan) using hexane:AcOEt (6:1) to (3:1). The purity of FX was confirmed by HPLC. Yield FX purity was approximately 99%. Tacrolimus was purchased from Cayman Chemicals (Ann Arbor, MI, USA).

### 4.2. Animals

10 week-old male Nc/Nga mice were purchased from SLC (Kyoto, Japan) and maintained under conventional conditions with fur mite infection. Mice showing a mild phenotype were divided into three groups: Vaseline control, Vaseline, or 0.1% FX containing Vaseline and 0.1% tacrolimus ointment (NIPRO, Osaka, Japan). Respective materials were applied to the back of mice after removing hairs (once a day, one fingertip unit) as described previously [20]. After five weeks, auricular lymph node dissection, blood, and skin were sampled for examination of gene expressions and histological analysis. TEWL of skin was measured at room temperature at 40% to 60% humidity with a Vavoscan AS-VT100RS (Asahi Biomed, Yokohama, Japan). A total clinical severity score for AD-like lesions was defined as the sum of the individual scores graded as 0 (none), 1 (mild), 2 (moderate) and 3 (severe) for each of five signs and symptoms (itch, erythema/hemorrhage, edema, excoriation/erosion and scaling/dryness) according to a slight modification of the criteria described previously [51]. Assessment was performed by an investigator who was blind to the grouping of the animals. Five week-old male NC/Nga mice with no AD pathology showed a TEWL value of 10–20 g/m^2^·hour. Eight to fifteen week-old C57BL6N mice and bone marrow cells were cultured in media containing WHEI-3 (National Institute of Biomedical Innovation: NIBI, Osaka, Japan) and conditioned medium to obtain mature bone marrow derived mast cells (see below). The protocol used here meets the guidelines of the Japanese Society for Pharmacology and was approved by the Committee for Ethical Use of Experimental Animals at Ritsumeikan (BKC2017-001).

### 4.3. Histological and Histocytochemical Analysis

Tissues were fixed with 10% (*w*/*v*) neutral buffered formalin solution and embedded in paraffin blocks. The blocks were sectioned by microtome (Leica Biosystems, Wetzlar, German) as described previously [52]. Section (5 μm thick) were stained with toluidine blue or congo red and examined by light microscopy (BX51 /DP21; Olympus, Tokyo, Japan). Sections were scanned with Olympus virtual slide scanning systems (Tokyo, Japan). The positive cells were counted using cellSens software (Olympus).

### 4.4. Quantitative Real-Time RT-PCR

Isolated skin tissues were homogenized and total RNAs were extracted using Sepasol^®^ (Nacalai Tesque, Kyoto, Japan) and synthesized in a cDNA pool as described previously [20]. Total RNAs preparation were also performed from six-well culture plates at confluency. Real-time RT-PCR analysis was performed as described previously [20]. The experiments were performed at five different cDNA pool dilutions. PCR products were normalized against Gapdh and measurements between samples were compared by cycling threshold (*C*t). Primer sequences used are summarized in Appendix A. A non-regulated housekeeping gene Gapdh served as an internal control and was used to normalize for differences in input RNA.

### 4.5. BMMCs Cultures

WEHI-3 cells were also cultured in RPMI1640 and supplemented with 10% FBS and antibiotics. For preparation of WEHI-3-CM, cells were centrifugated and supernatants were filtered by 0.2 μm filter. Murine bone marrow cells were cultured in PRMI1640-based VEHI-3-CM supplemented with 10% FBS and antibiotics as according to the protocol described previously [53].

### 4.6. Immunocytochemical and Immunohistochemical Analysis

BMMCs were plated to the slide and were fixed by heating with 10% (*w*/*v*) neutral buffered formalin solution. Cells were reacted with mouse anti-mast cell tryptase antibody (AB2378, Abcam, Cambridge, UK). Cells were then reacted with FITC-conjugated anti-mouse IgG (Santa Cruz Biotechnology, Inc., Santa Cruz, CA, USA) and 1µg/mL DAPI (4′,6-diamidino-2-phenylindole dihydrochloride). Paraffin sections were reacted with respective antibodies and analyzed as described previously [53]. Sections were subjected to immunohistochemistry using antibodies of Alexa Fluor 488^®^-anti-mouse Il-10 (#505013), Brilliant Violet 421™-anti-mouse TGF-β1 (#141407), Pacific Blue™-anti-mouse IL-2 (#503820), Alexa Fluor^®^ 488-anti-GATA3 (#653808), PE-anti-mouse Ly-6A/E (Sca-1) (#108108), PE-anti-mouse IL-33Rα (ST2) (#146608) and PE-anti-mouse/human IL-5 (#504304) (BioLegend, San Diego, CA, USA), FITC-anti-mouse/human CD45R (#11-0452-82) (Thermo Fisher Scientific, Waltham, MA, USA), PE-anti-mouse CD25 (#130-120-766) (Miltenyi Biotech, Bergisch Gladbach, Germany). All antibodies were used at a 1:500 dilution.

### 4.7. Statistical Analysis

Data are expressed as mean ± SEM. Significance was tested using Student’s t-test or, where multiple comparisons were required, One-way analysis of variance (ANOVA) was used with post hoc Bonferroni test. A *p*-value of less than 0.05 was considered to be significant.

## Figures and Tables

**Figure 1 ijms-21-02180-f001:**
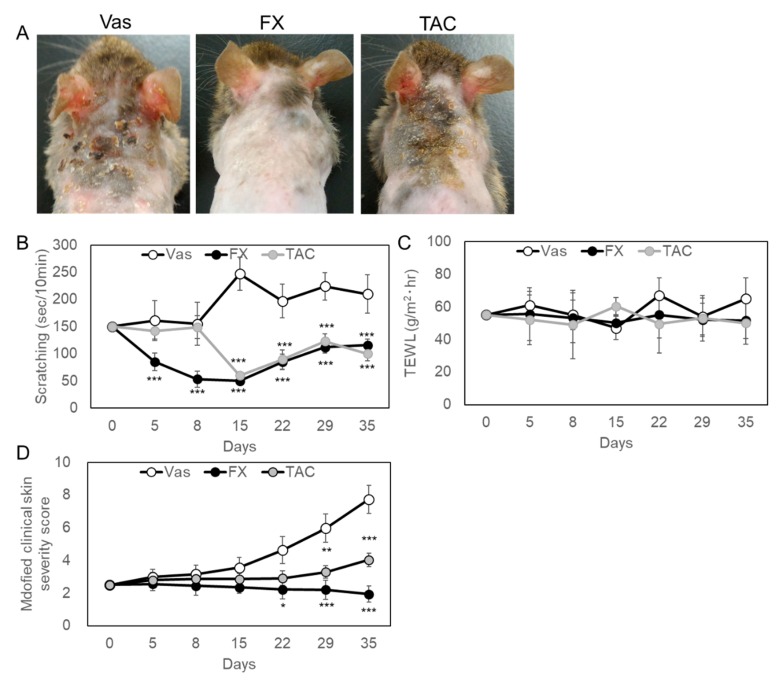
Feature of typical experimental findings in fucoxanthin (FX) or tacrolimus (TAC) treated Nc/Nga mice. Vaseline (Vas), Vas containing FX or TAC were applied to shaved back skin daily for five weeks. Shaving was performed every two weeks. (**A**) Mice skin symptoms after five weeks of application. Each topical application was treated to groups of six mice and the representative symptoms were shown after five weeks. (**B**) The scratching behavior was inhibited by FX- or TAC-treated groups compared with Vas-treated group. The scratching behavior was videotaped in the morning after the indicated number of days from the start of the test and scratching time was measured with the hind legs every 10 min. (**C**) TWEL values were not influenced by FX or TAC. (**D**) Clinical skin condition in FX or TAC treated skin. Atopic dermatitis (AD) lesions were scored with slight modification of the human criteria. Values are means ± SEM. One-way ANOVA followed by a Bonferroni post hoc test. * *p* < 0.05, ** *p* < 0.01, *** *p* < 0.005 compared with Vas control.

**Figure 2 ijms-21-02180-f002:**
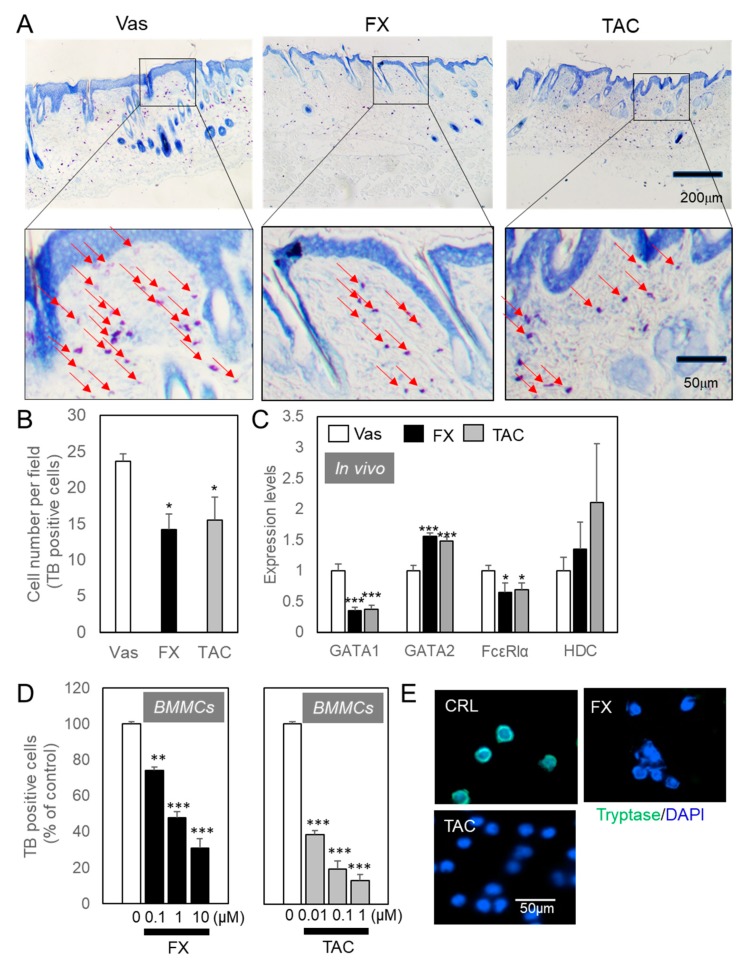
FX or TAC inhibited mast cells formation. After five weeks of treatment, each tissue was excised from the occipital region to around the scapula. Paraffin sections were stained with TB. Sections prepared from all mice were scanned and the number of TB-positive cells was counted. From each section, 30 visual fields were measured. The representative images are shown in (**A**) and a comparison of the number of mast cells per unit field is shown in (**B**). Arrows indicate TB-positive cells. (**C**) FX or TAC inhibited mast cells differentiation in vivo. Skin samples were isolated after five weeks of being treated with respective compounds (*n* = 6). mRNA expression levels were analyzed by qRT-PCR. (**D**) FX or TAC inhibited TB-positive bone marrow-derived mast cells (BMMCs) formation in a concentration-dependent manner. After three weeks, cells were fixed and stained. BMMCs were cultured with VEHI-3-CM in the presence or absence of FX or TAC for three weeks. Medium change was performed every other day. Data are shown as percent of dimethyl sulfoxide (DMSO) control (*n* = 6). Values are means ± SEM. * *p* < 0.05, ** *p* < 0.01, *** *p* < 0.005 compared with Vas or DMSO control. (**E**) The effect of FX or TAC on tryptase-positive granules formation in BMMCs. Immunocytochemical analysis was performed after three weeks of treatment with 1 μM FX or 0.1 µM TAC. Tryptase-positive reaction (green) were not observed in FX- or TAC-treated cells compared to DMSO control. Cells were counter-stained by 4’,6-diamidino-2-phenylindole (DAPI) (blue).

**Figure 3 ijms-21-02180-f003:**
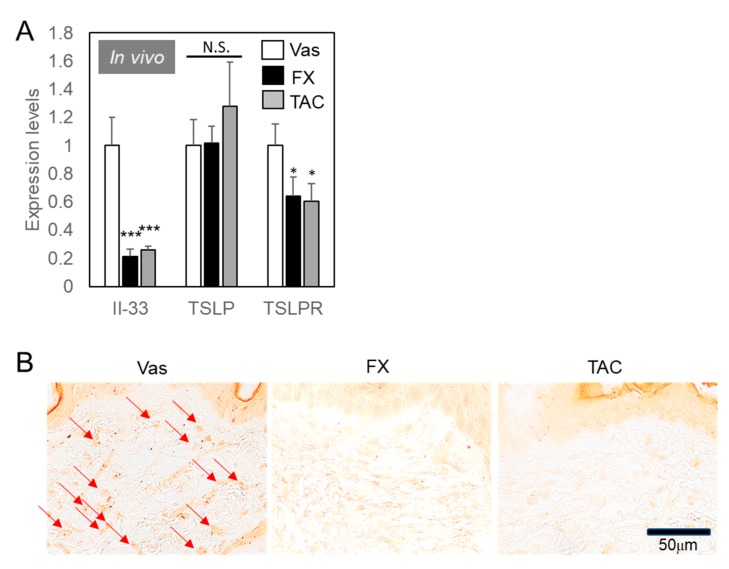
Il-33-dependent eosinophil recruitments blocked by FX or TAC. (**A**) Keratinocyte-derived cytokines regulated by FX or TAC. Samples used in Figure 2C were analyzed using specific primer pairs. Values indicated means ± SEM from individual mouse averages (*n* = 6) * *p* < 0.05, *** *p* < 0.005 compared with Vas control. N.S. = not significant. (**B**) Eosinophil recruitments were blocked by FX or TAC. Specimens were reacted with 0.5% congo red solution and washed. Arrows indicate congo red-positive eosinophils.

**Figure 4 ijms-21-02180-f004:**
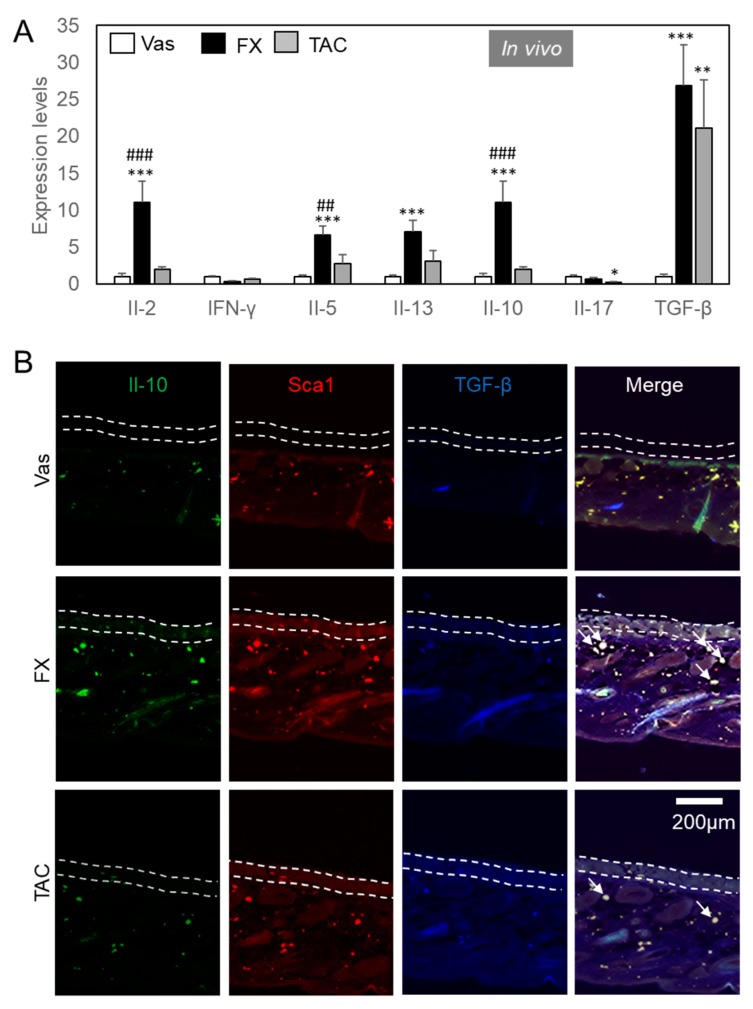
FX stimulated Il-10^+^TGF-β^+^Sca1^+^ ILCreg. (**A**) Cytokine expression levels regulated by FX or TAC. Samples used in Figure 3 were analyzed using specific primer pairs. Values are means ± SEM. * *p* < 0.05, ** *p* < 0.01, *** *p* < 0.005 vs Vas, ## *p* < 0.01, ### *p* < 0.005 vs TAC. (**B**) Il-10^+^TGF-β^+^Sca1^+^ ILCreg dominantly expressed in FX-treated Nc/Nga mice. Sections were reacted with Alexa Fluor 488^®^ anti-mouse Il-10, Brilliant Violet 421 ™ anti-mouse TGF-β1, and PE anti-mouse Sca1 antibodies. All antibodies were used at a 1:500 dilution. The area between the dot lines indicates the epidermal layer. Arrows indicate typical triple-positive cells.

**Figure 5 ijms-21-02180-f005:**
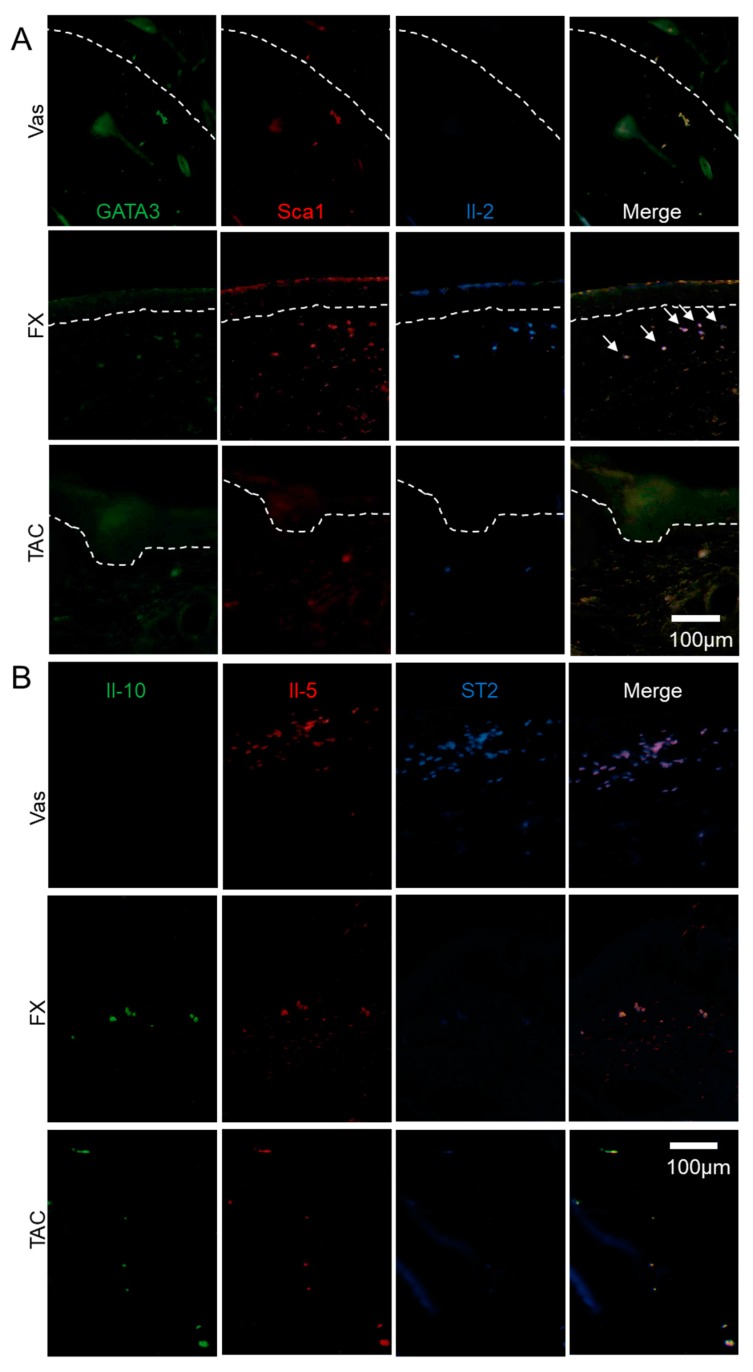
The localization of ILC2 and ILCreg modulated by FX or TAC. FX stimulated Il-2^+^GATA3^+^Sca1^+^ ILCreg. Il-2^+^GATA3^+^Sca1^+^ ILCreg dominantly expressed in FX-treated Nc/Nga mice (**A**) and Il-5^+^ST2^+^ ILC2 in the Vas treated group (**B**). Sections were reacted with respective fluorescence-labelled antibodies. The area between the dotted lines indicates the epidermal layer. Arrows indicates typical GATA3^low^Il-2^high^ cells (A). Panel (B) is focused on the dermis.

**Figure 6 ijms-21-02180-f006:**
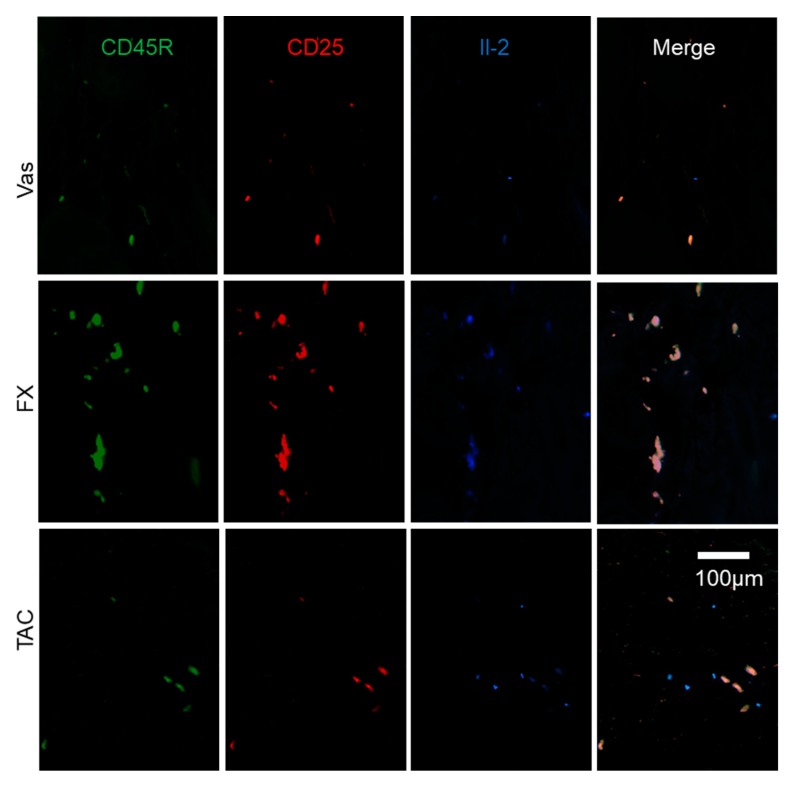
The characterization of Il-2 producers regulated by FX and TAC. FX stimulated Il-2 production in CD45R^+^CD25^+^ ILCregs. Some Il-2^+^CD45R^+^CD25^+^ ILCregs were observed in TAC treated skin and others were CD45R^-^CD25^-^ cells. The panel is focused on the dermis.

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
