# Peer review of "Fucoxanthin Ameliorates Atopic Dermatitis Symptoms by Regulating Keratinocytes and Regulatory Innate Lymphoid Cells"

_ijms, 2020, doi:10.3390/ijms21062180_

Round 1

Reviewer 1 Report

  1. TAC is often used for triamcinolone. It may be better to spell tacrolimus out to avoid confusion.
  2. I don’t know what “AD is a major skin barrier disorder that promotes the onset” means
  3. One of the key causes of poor response in AD is poor adherence; I’m not sure we know that tacrolimus does not work when it is actually used.
  4. Phrases like “have been found to have” can be shortened to “have”. Phrases with known, shown, found, demonstrated, proven, etc, can be deleted, because if something has been shown, it just is.
  5. Similarly, phrases ending in “that” like “it was reported that”, “we previously reported that”, and “Recent studies demonstrated that” can be eliminated for the same reason.
  6. It was not clear to me how the authors can conclude that the reduction in itch is due to modulating mast cells. To do that, one would have to get rid of the mast cells or prevent mast cells from experiencing the fucoxanthin effects.  All we know is that itch goes down and cytokines change; the type of study that was done does not allow us to determine what is cause and what is effect.

Author Response

Thank you for the thoughtful and constructive feedback you provided regarding our manuscript, entitled " Fucoxanthin ameliorates atopic dermatitis symptoms by regulating keratinocytes and regulatory innate lymphoid cells." in International Journal of Molecular Sciences.

We agree with the reviewer’s concern and have provided brief response to the reviewer’s concern below. Modifications to the revised text are in red to facilitate the re-review process. We hope that the all changes made to the manuscript meet your satisfaction. The manuscript substantially improved after the author's revision and we note with pleasure that the peer-review system is useful in the direction of publishing better papers.

  1. We changed the title.
  2. We added the data of modified clinical skin severity in Figure 1D.
  3. We added the data of inflammatory cytokine expression levels and toxicity tests of FX in supplementary figures.
  4. The manuscript has been proofread by native.

As you pointed out, we cannot conclude that FX acts on mast cells and has improved AD symptoms. We concluded that FX mainly affect ILCreg, consequently. The abstract was also modified because a common mechanism with tacrolimus may be anti-inflammatory through keratinocytes.

Reviewer 2 Report

Natsume et al present a paper that investigated the effect of fucoxanthin (Fx) on atopic dermatitis (AD) symptoms in a mouse model of dust mite sensitized NC/Nga mice. It was shown that 0.1% Fx applied topically significantly improved atopic dermatitis symptoms as compared to the vehicle vaseline. The effect was similiar to tacrolimus (tac) ointment.

Immunohistological and immunological investigations were performed on skin biopsies, lymph nodes and bone marrow derived cells. The authors conclude that the improvement of AD was mainly mediated by regulating mast cells and regulatory innate lymphoid cells.

I have some comments and suggestions to improve the paper.

In the abstract it is postulated that FX topical treatment dramatically ameliorated skin atopic symptoms and itching behavior over Tacrolimus. The data shown do not support this statement. Fx is superior to the vehicle and it improves itch more rapidly than tac (already after 5 days), but at 2 weeks there is no difference between tac and fx.

Also, it has not been systematically tested that fx dramatically ameliorated (clinical) skin atopic symptoms. A clinical skin scoring of AD symptoms (i.e. a kind of SCORAD) is missing which could have easily been monitored.

Toxicity testing is missing. Previously the toxicity of Fx on the skin was tested in a skin equivalent. The data have just been published in: Pharmaceutics. 2020 Feb 5;12(2). pii: E136. doi:10.3390/pharmaceutics12020136. This paper should be cited. Additionally, toxicity testing (i.e. LDH release, ATP metabolism) of Fx in the cell culture models used and in keratinocytes could easily be performed and be included in the data presentation.

Minor:

The concentration of tacrolimus should be indicated (probably 0.1%) as well as the tacrolimus vehicle. If a commercially available procuct was tested please indicate the manufacturer.

English spelling errors and phrasing should be corrected by a native speaker.

Author Response

Thank you for the thoughtful and constructive feedback you provided regarding our manuscript, entitled " Fucoxanthin ameliorates atopic dermatitis symptoms by regulating keratinocytes and regulatory innate lymphoid cells." in International Journal of Molecular Sciences.

We agree with the reviewer’s concern and have provided brief response to the reviewer’s concern below. Modifications to the revised text are in red to facilitate the re-review process. We hope that the all changes made to the manuscript meet your satisfaction. The manuscript substantially improved after the author's revision and we note with pleasure that the peer-review system is useful in the direction of publishing better papers.

  1. We changed the title.
  2. We added the data of modified clinical skin severity in Figure 1D.
  3. We added the data of inflammatory cytokine expression levels and toxicity tests of FX in supplementary figures.
  4. The manuscript has been proofread by native.

Thank you for your very valuable comments. By scoring the skin symptoms, we were able to focus on the main effects of FX. In addition, by performing a cytotoxicity test using FX, I was able to notice new effects of FX.

Round 2

Reviewer 2 Report

All issues have been adequately adressed by the authors. This is a fine paper and I suggest to accept it as it stands